# Layer Morphology and Ink Compatibility of Silver Nanoparticle Inkjet Inks for Near-Infrared Sintering

**DOI:** 10.3390/nano10050892

**Published:** 2020-05-07

**Authors:** Dieter Reenaers, Wouter Marchal, Ianto Biesmans, Philippe Nivelle, Jan D’Haen, Wim Deferme

**Affiliations:** 1Institute for Materials Research (IMO), Universiteit Hasselt, Wetenschapspark 1, B-3590 Diepenbeek, Belgium; dieter.reenaers@uhasselt.be (D.R.); Wouter.Marchal@uhasselt.be (W.M.); Ianto.biesmans@uhasselt.be (I.B.); philippe.nivelle@uhasselt.be (P.N.); Jan.Dhaen@uhasselt.be (J.D.); 2IMEC vzw, Division IMOMEC, Wetenschapspark 1, B-3590 Diepenbeek, Belgium

**Keywords:** near-infrared, photonic sintering, silver, nano particle, inkjet, printing, light absorption

## Abstract

The field of printed electronics is rapidly evolving, producing low cost applications with enhanced performances with transparent, stretchable properties and higher reliability. Due to the versatility of printed electronics, industry can consider the implementation of electronics in a way which was never possible before. However, a post-processing step to achieve conductive structures—known as sintering—limits the production ease and speed of printed electronics. This study addresses the issues related to fast sintering without scarifying important properties such as conductivity and surface roughness. A drop-on-demand inkjet printer is employed to deposit silver nanoparticle-based inks. The post-processing time of these inks is reduced by replacing the conventional oven sintering procedure with the state-of-the-art method, named near-infrared sintering. By doing so, the post-processing time shortens from 30–60 min to 6–8 s. Furthermore, the maximum substrate temperature during sintering is reduced from 200 °C to 120 °C. Based on the results of this study, one can conclude that near-infrared sintering is a ready-to-industrialize post-processing method for the production of printed electronics, capable of sintering inks at high speed, low temperature and with low complexity. Furthermore, it becomes clear that ink optimization plays an important role in processing inkjet printable inks, especially after being near-infrared sintered.

## 1. Introduction

Over the last decade, drop on demand (DoD) inkjet printing has become a widely used technique to print thin conductive silver patterns for various applications including: Radio-frequency identification (RFID) tags, temperature and humidity sensors, flexible circuits, heaters, strain sensors, capacitive sensors, on many types of substrates like glass, plastics, paper and textiles [1,2,3,4]. DoD inkjet printing combines low manufacturing cost, low material waste, reduction of inventory with high resolution and fast prototyping [5]. Since inkjet printing requires only software modifications in contrast to hardware modifications for screen printing, slot die printing, gravure printing and offset printing [6,7,8,9], it creates a great improvement in patterning flexibility. Inkjet printing can be performed with different types of inks. For printed electronics, the most applied ink types are metal nanoparticle inks [10], metal-organic decomposition inks (MOD) [11,12], polymer inks and (graphene) micro flake inks [13,14]. In this study, only metal nanoparticles are studied. Inkjet printing of patterns with nanoparticle inks should be followed by a curing step in order to achieve conductive patterns. In general, the curing process consists of the evaporation of solvents, followed by the removal of stabilizing agents and finally the sintering of the nanoparticles to form a conductive layer [10].

In recent literature, numerous methods to cure thin silver layers have been investigated and described for industrial scale applications. The most commonly used curing method is thermal sintering, where heat is applied with an oven or a hotplate. Depending on the ink composition, this method requires high temperatures of 140 to 400 °C for 10 to 60 min to achieve a conductive printed silver layer [10,15,16,17]. However, these high temperatures and long curing times make it impossible to sinter silver patterns on heat-sensitive substrates without causing thermal degradation [18]. Nevertheless, this sintering method results in very smooth surfaces (Ra: 10–20 nm) with a low sheet resistance, ranging between 0.04 and 0.13 Ω/□ for commercially available JS-B40G silver nanoparticle ink, bought from Novacentrix (Austin, TX, USA), as one of the inks studied in this paper [10]. To overcome the thermal degradation effects of substrates during curing, many selective sintering approaches are investigated in recent literature [19,20,21,22]. Electrical, chemical and radiation-induced sintering methods are considered as promising alternatives to remediate the abovementioned detrimental thermal effects.

The onset of electrical sintering is achieved by preheating the sample until it is slightly conductive, followed by the application of a fixed current. Due to the Joule heating, the applied current generates heat and, hence, sintering. As the sintering proceeds, the conductivity of the samples increases, reducing the Joule-effect inside the ink layer and thus less heat is generated, which naturally halts the sintering further. However, the need for physical contact between the probes and the sample and the possibility of local overcurrent damages the printed structure and therefore seems to be inadequate for roll-to-roll printing applications. In case of chemical sintering, a sintering agent is added to the ink, and a reactive decomposition of the dispersing and/or capping agent will take place. Hence, the nanoparticles will be forced to fuse into a conductive path. Finally, radiation-induced sintering can be segmented into laser-, microwave-, infrared (IR)-, ultraviolet (UV) and near-infrared (NIR) sintering. Laser sintering uses precisely targeted high-power light pulses of 8 picoseconds to 1 s. These short pulses demand expensive equipment and complicated handling, therefore this method is difficult and less attractive to use in roll-to-roll printing [19,20,21,22]. Compared to thermal sintering and laser sintering, microwave sintering achieves better sintering speeds. Although microwaves ensure rapid heating and sintering, with lower energy consumption than oven sintering and its compatibility with other sintering techniques, it is slightly less selective compared to laser sintering because the microwaves could heat the whole sample depending on its absorption coefficient of the substrate [23,24,25]. The inception of UV sintering started with an ingenious combination of radiation-induced and chemical sintering where a photo initiator added to the ink will decompose under UV exposure resulting in a reaction. Finally, a short thermal treatment is performed to form a conductive pathway [26].

Infrared (IR) sintering is a relatively new sintering technique. It combines less process complexity, with shorter processing times [27,28]. Since it heats up the whole sample, this technique is less selective than laser sintering for most substrates. Due to the high absorption coefficient of silver nanoparticle inks in the NIR spectrum, NIR radiation will interact efficiently with the silver nanoparticles resulting in very local and fast heating. Meanwhile, most of the substrates remain relatively unaffected since it has a much lower absorption coefficient in the NIR spectrum [29]. Cherrington et al. proved that NIR sintering could be a fast alternative for thermal sintering and that it can also be applied for flexible polymer substrates in roll-to-roll applications [16].

It has been shown before that for NIR sintering, there is a promising synergy of short sintering durations, compatibility with heat-sensitive substrates, process simplicity and selectivity. Therefore, this study looks deeper into this sintering method, focusing on roll-to-roll industrialization without having to cut down on important properties such as conductivity and surface roughness. At first, a comparison of the direct influence of NIR sintering relative to oven sintering is made. In addition, a more optimized ink towards NIR sintering is selected and investigated, while trying to reduce the maximum processing temperature, without losing the focus on roll-to-roll industrialization. An innovative way to monitor the moment the structure becomes conductive, based on the emissivity of the printed layer, is applied and compared to standard four-probe sheet resistance measurements.

## 2. Materials and Methods

### 2.1. Sample Preparation

Two different silver nanoparticle inks (AgNP); JSB-40G and JSA-102A (both 40 wt.% loading) were purchased from NovaCentrix (400 Parker Dr #1110, Austin, TX, 78728, USA). Both inks are designed specifically for different sintering methods: JSB-40G requires thermal sintering with a temperature of at least 180 °C, whereas JSA-102A is designed for photonic sintering [17,30]. Glass slides (26 × 25 mm) were ultrasonically cleaned in isopropanol for 15 min and dried with a dust free Amplitude Sigma alpha wipe (Contec, Spartanburg, SC, USA), and subsequently dried under nitrogen. Both Ag NP inks were printed separately in an 8 by 8 mm square pattern on the precleaned glass substrates by a Fujifilm Dimatix DMP 2831 DOD inkjet printer (Fujifilm, Minato, Tokyo, Japan), using 10 pl cartridges (20 µm drop spacing, nozzle voltage 24–26 V, jetting frequency 5 kHz). Printing is followed by NIR sintering. The NIR sintering setup, visible in Figure 1, consists of a 400 W NIR light source unit with a peak intensity wavelength of ±950 nm (AdPhos Innovative Technologies GmbH, Bruckmühl-Heufeld, Germany), and a home-built height adjustable PTFE sample holder with two embedded thermocouples. A custom-made LabVIEW 2017 (National Instruments, Austin, TX, USA) program was controlling the power delivery from the NIR source. Depending on the requirements, the versatile NIR sintering setup could deliver radiation in continuous or flashing mode. During sintering, the temperature was logged using the picolog TC-08 software (Pico Technology, Saint Neots, UK) and the lamp to sample distance is varied from 0.5 to 3 cm. 

### 2.2. Device Characterization

The sheet resistance of the sintered device was measured with a home-built Van der Pauw measurement system. Thickness and surface roughness of the ink layer were determined by profilometry (Brucker Dektak XT, ISO4287). A cut-off bandpass filter (0.8 mm) was used to filter out the waviness from the Ra measurements. Scanning electron microscopy (SEM) (FEI Quanta 200F, Hillsboro, OR, USA) was also performed to get deeper insights on the grain size and particle sintering. To measure the emissivity transition, related to the nanoparticle sintering to continuous conductive features, the samples were taken out of the NIR sintering setup at different timings and placed on a hotplate to stabilize their temperature at 45 °C. Subsequently a FLIR E40 handheld thermal camera was applied to measure the average and spot emissivity of the printed surface; the hotplate, reflection and environment temperature were taken into account. The average emissivity is measured over the full printed area of 8 by 8 mm, the spot emissivity is measured at the Van der Pauw probe location. DLS experiments were performed on a Malvern zetasizer nanoseries (Malvern Panalytical, Malvern, UK) device using a 532 nm wavelength. Dispersions were diluted in ethanol. UV-Vis measurements were conducted with the Cary 5000 UV-Vis-NIR spectrometer (Agilent technologies, Santa Clara, CA, USA) in scan mode in clear face quartz cuvettes from 250 up to 1300 nm with a 1 nm resolution on a diluted ink sample (in ethanol).

## 3. Results and Discussions

First, a comparison between oven sintering and NIR-continuous sintering is made, with focus on sintering duration, sheet resistance (conductivity) and morphology. Then, the NIR absorption of the ink is studied based on thermal imaging. A correlation is found in emissivity and sheet resistance. Thereafter, the importance of ink optimization is illustrated. Crucial ink properties are discussed and assigned. Based on these findings, NIR flash sintering is introduced and experiments were performed to pinpoint the best sintering conditions, taking into account the trade-off between sintering time, temperature, conductivity, morphology.

### 3.1. Oven Sintering Versus NIR-Continuous Sintering

To investigate the potential improvement by NIR sintering compared to oven sintering, the sintering temperature of the ink-covered sample loci are kept constant at 200 °C for both sintering methods. Although it is possible to sinter this ink at both lower and higher temperatures, 200 °C is chosen, as this temperature is a good trade-off between required sintering duration and temperature, as indicated by the ink supplier. Having a lower sintering temperature (180 °C is indicated as the lowest possible temperature to achieve sufficient sintering for decent conductivity) will increase the time for sintering, whereas higher sintering temperatures will affect layer formation in a negative way. At the same time, the uncovered substrate temperature is variable and rising continuously in the course of the sintering procedure. The increase of substrate temperature during sintering is caused by the intensification of NIR light by the PID (Proportional–Integral–Derivative) controller. The PID controller intensifies the NIR light since the printed ink on the substrate will progressively absorb less NIR radiation during sintering, and it is designed to maintain the temperature of the printed area constant at 200 °C [16]. Nevertheless, the surrounding substrate temperature remained well below 200 °C.

Figure 2 shows the sheet resistance as a function of sintering time for both oven sintering (black) and NIR-continuous sintering of the JS-B40G (blue-red-green-purple) ink at 200 °C.

During NIR sintering, it is shown that the conductivity was slightly improved by decreasing the sintering distance. Increasing the sintering distance resulted in an increased lamp intensity required to reach 200 °C and vice versa. Consequently, the NIR radiation per square centimeter remained unchanged because of the PID controller. However, changing the lamp intensity will slightly influence the NIR wavelength and therefore it can somewhat influence the sintering behavior which could explain the change in conductivity between different sintering distances. Additionally, decreasing the sintering distance results in a faster heat up time, causing relatively better conductivity at shorter sintering distances. Figure 2 also illustrates a significant decrease in sintering time from 30–90 min to 2–5 min when switching from thermal to NIR sintering, in accordance with earlier studies by J. Perelaer [23]. However, the reduction in sintering time resulted in a slight increase in sheet resistance, from 0.027–0.050 Ω/□ to 0.055–0.085 Ω/□. Preheating the printed samples at 150 °C for 10 min in a hot oven, could reduce the sintering time with NIR down to 30 s, but it will prolong the total processing time and again increase the sheet resistance to 0.12–0.19 Ω/□. A deeper look into the mechanisms causing this significant decrease in sintering time compared to oven sintering is shown in Section 3.3.

Morphology and surface roughness play an important role in ensuring the durability and flexibility for possible future application [31]. It is evident from Figure 3A that oven sintered samples already attain their optimal morphology from 30 min processing time onward (the point where the silver depositions start showing electric conductivity).

The average Ra roughness will not decrease upon extended heat treatments and varies around 13.5 nm, as can be seen in Figure 3A. When sintering times are much shorter than 30 min, SEM images reveal that most nanoparticles are still in their original shape, not forming continuous tracks (Figure 2). Upon extended sintering times, the nanoparticles coalesce together, creating a smoother conductive surface. NIR-continuous sintering also results in a Ra roughness which will remain constant once conductivity is obtained and sintering is fulfilled. However, when comparing surface roughness after sintering at different distances, presented in blue (Figure 3A), an optimum Ra roughness of about 25 nm can be observed at a sintering distance between 1 and 2 cm. Sintering at too short distances will result in an intense and quick heating rate, causing searing in the top layer and creating an irregular surface while sintering at too long a distance will lead to an intensity which is too low to ensure good particle migration and coalescence for this specific ink, because the maximum lamp power will be reached, again resulting in a rough surface and a correlated higher sheet resistance.

Furthermore, the coffee ring effect was observed during sintering with both methods. As mentioned before, the JSB-40G ink is not optimized for NIR sintering. As such, the coffee-ring effect is expected to manifest itself due to the non-optimized gradient of evaporation rates over the ink layer surface. At the edge of the ink surface, the evaporation rate of liquid is higher due to the bigger surface area versus bulk material, hence creating an outward convective flow (capillary flow) in the liquid ink state upon heating. This stream of liquid carries nanoparticles to the edge [32]. The opposing flow (Marangoni flow), in this case, is less prominent, resulting in an imbalance of particle migration. Consequently, silver nanoparticles build up at the edge and are immobilized after solvent evaporation, resulting in a crater-like deposition pattern. These coffee rings reduce layer quality and can potentially lower the resolution of printed applications. JSB-B40G is an ink optimized for oven sintering and therefore it’s engineered to have a low tendency to create these aforementioned flows in a thermal convection oven. This leads to a coffee ring which is 130% to 140% of the regular layer thickness. However, in NIR sintering, heat is transferred through radiation instead of thermal convection. This heating process has a different influence on these convective flows in the ink, creating an imbalance. In Figure 3B, the average layer and coffee ring thickness are presented. NIR-continuous sintering at 1 cm creates a coffee ring which has a thickness of about 240% of the regular layer thickness. At 3 cm, the coffee ring is very significant, raised towards 350% of the normal layer thickness

### 3.2. Silver Nanoparticle Ink’s Emissivity and NIR Absorption

To justify the decrease in cut-off sheet-resistance upon increasing sintering duration, the emissivity and absorbance of the inks are investigated in detail. Cherrington et al. [16] have shown that there is a transition in light absorbance before and after curing, Tobjörk et al. also stated that the decrease of absorbance during sintering acts as natural protection against overheating during photonic sintering [33]. However, one has never visualized this transition directly. This transition also corresponds with the abovementioned phenomenon of gradually increasing NIR radiation output as the PID controller was set to maintain a constant temperature on the ink-covered surface.

Before curing, the JS-B40G ink has an emissivity (absorptivity) of ±0.96. It can be seen from Figure 4A that there is a rapid decrease in emissivity for the NIR sintered JS-B40G between 0 and 120 s and a more gradual decrease after 120 s, eventually attaining a value of 0.02. The rapid decrease in average emissivity is in accordance with the results shown by Cherrington et al. and it indicates the occurrence of a series of events associated with the curing process like; solvent evaporation, the release of capping agent and particles size growth [16]. The latter process is clearly noticeable in the SEM images presented in Figure 2. As the size of the particles increases, the optical density, as well as the emissivity, will decrease [34]. The small residual decrease after 120 s is related to the fact that the pattern edges sinter slower and therefore still have a higher emissivity than the center, which influences the average emissivity measurement. This effect is visible in Figure 5. Adding the sheet resistance to Figure 4A, after different sintering durations, clarifies that conductivity remains constant when emissivity is low. When finally considering the spot emissivity measurement in Figure 4A, which was located at the contact points for sheet resistance measurements to exclude the edge effect, an even stronger relation between emissivity and sheet resistance is observed.

To provide additional support to the correlation between emissivity and sheet resistance, the same test was performed for oven sintered JS-B40G samples which needed a much longer curing time, as demonstrated before (Figure 2). Again, Figure 4B shows a strong relation between average/spot emissivity and sheet resistance. It is clear that the emissivity transition is a result of the sintering step during the curing process. The only difference in emissivity progression of NIR versus oven sintered samples is the emissivity transition pattern. For NIR sintered samples, emissivity starts of lower in the center. This is caused by the heat sink effect of the substrate. The glass substrate absorbs less NIR radiation and thus stays relatively unaffected and cool, meanwhile the ink layer absorbs a high percentage of the radiation and quickly heats up, creating a considerable temperature gradient. The surrounding glass substrate retracts heat out of the edges of the silver pattern; therefore, the ink edges will remain slightly colder than the center and thus sinter at a slower rate. On the contrary, heat is homogeneously distributed over the whole sample during oven sintering, resulting in a homogeneous temperature profile imposing the same sintering speed over the sample. However, some slight inhomogeneous emissivity can be observed in Figure 5 (20 and 30 min), caused by the difference in ink layer thickness due to the printing process and coffee ring formation. Locations where the ink layer is thicker require more time to sinter than locations where the ink layer is fairly thinner.

### 3.3. Importance of Ink Optimization for NIR Sintering

The direct improvement of NIR sintering duration versus oven sintering, presented in Figure 2, lies in the local heating effect. Vandevenne et al. mimicked the local temperature with oven sintering and showed the evolvement of the different sintering steps by performing atomic force microscope (AFM) measurements after heating in an oven at different temperatures up to 400 °C [10]. It could be observed that the local temperature needs to go well above 200 °C to achieve quick sintering and necking of the nanoparticles to form bigger (clustered) particles optimizing the percolation path. In a well calibrated oven set at a temperature of 200 °C, the local temperature is limited to this setpoint value, leading to a longer required sintering duration. For NIR sintering however, as presented in Figure 6, the heat absorption of the nanoparticles according to their specific absorption spectrum is more efficient. Hence, the local heating effect will not only include higher temperatures but also the overall temperature of the layer and substrate remains lower as compared to oven sintering (as there is local heating and direct sintering), implying no (or less) spread of the heat. Even though the required sintering time for the JS-B40G ink is significantly reduced by switching from thermal to NIR curing, the process is still not roll-to-roll compatible for printable electronics. In addition, thermal degradation of conventional plastic substrates (such as Polyethylene terephthalate (PET)) cannot be eliminated since temperatures of 200 °C are still required to successfully sinter the silver patterns. Nevertheless, the degradation effects will be reduced due to the significant shorter sintering time, but optimization of the ink is clearly required. A fundamental insight regarding the different mechanisms such as particle migration (coalescence), aggregation and Ostwald ripening, during sintering is necessary to optimize the ink [35,36,37,38,39,40]. Ostwald ripening can occur at the early stage of sintering when particles still have their original size and are driven by the Gibbs–Thompson phenomenon. If the ink consists of a widespread nanoparticle size distribution, then smaller particles agglomerate into bigger particles, as smaller particles are less stable due to high surface energy [35,36]. As sintering progresses, the surface free energy of all particles will gradually reduce, resulting in less Ostwald ripening. Meanwhile, particle migration (coalescence) becomes the main sintering mechanism. The bigger particles will coalesce and aggregate together randomly, forming necks and eventually completely merge together. Besides these main mechanisms, there are still some particle and ink formulation related properties which could influence the sintering progress. One of these is the effect of a polymer capping agent used to stop nanoparticles from agglomerating before sintering. As the thickness of the capping layer is less than a few nm and decompose in the primary stage of the sintering procedure, its effect on sintering is negligible [26,36]. However, the thermal stability of the capping plays a significant role on the sintering temperature required to form a conductive path. Another condition which affects the sintering is the type of solvent used in the ink. A low boiling point solvent evaporates much faster than a high boiling point solvent, resulting in the particles coming closer to each other faster and therefore help the two earlier described mechanisms (Ostwald ripening and coalescence). This has a significant influence on the sintering time. Finally, the last described condition is the particle size. Since the size of these nanoparticles is in the lower nm range (between 20 and 100 nm), they are subjected to scattering when being photonically sintered. Upon photo incidence, the localized surface plasmon resonance (LSPR) of the silver nanoparticles prevails if the incident light resonantly matches with the localized field of AgNPs and hence, alters the absorbance by scattering the radiation. LSPR primarily depends on the size of the AgNp, dielectric constant of solvents used and the wavelength of incident radiation. Figure 6 shows both ink absorbance and the lamp spectral radiation as a function of the wavelength.

The UV-VIS measurement shows that the JS-B40G ink has its maximum absorbance peak at 486 nm. Dynamic light scattering measurement of the ink batch undertaken by the supplier shows that the JS-B40G ink has an average particle size of 69 nm. When calculating the maximum extinction based on the Mie theory, 69 nm diameter particles have their maximum absorbance at 482 nm, which is in accordance with our experiment [41,42,43,44,45].

Increasing the particle size shifts the maximum absorbance peak towards a higher wavelength and matching better with the wavelength of our NIR light source. However, a bigger particle size will also reduce the maximum absorbance of the ink and reduce the effect of Ostwald ripening because these particles are more stable. This results in a trade-off system between the sintering mechanism speed and the amount of light, and thus heat absorbance.

Based on these parameters, a second commercially available ink was studied, namely, JS-A102A. This ink is stated to be optimized for photonic sintering. JS-A102A has an average particle size of 36 nm, based on a dynamic light scattering measurement (DLS) of the ink batch performed by the supplier. However, the size distribution in this ink is different from JS-B40G ink. DLS data show two Gaussian curves, instead of one for JS-B40G, with maxima at a particle diameter of ±12 nm and ±76 nm. The particle size distribution consists of both small and bigger particles, the maximum absorbance wavelength of bigger sized particles matches with the NIR radiation, increasing the absorption and meanwhile, the smaller particles facilitate the Ostwald ripening process. The consequence of Ostwald ripening and increase in absorption can accelerate the sintering process. Along with these two processes, the solvent composition also determines the optimization of ink for the sintering process. Thermogravimetric analysis (TGA) measurements, presented in Figure 7, illustrate a more gradual weight loss for the JS-B40G ink compared to JS-A102A, requiring higher temperatures for the complete removal of organic, conductivity-impeding ink components. The gradual weight loss before 230 °C in JS-B40G can be related to the evaporation of multiple high-boiling solvents such as diethylene glycol monobutyl ether and reveals a complex solvent formulation blend. On the other hand, JS-A102A, only consists of a less complex, lower boiling point solvent combination: the peak in the differential thermogravimetry (DTG) curve is situated around 130 °C, and the mass loss is more concentrated compared to the JS-B40G ink. This modification reduces the maximum sintering temperature and also the sintering duration.

### 3.4. NIR Flash Sintering of JS-A102A Ink

From the TGA measurement, it can be inferred that the JS-A102A ink shows the most promise to be combined with photonic sintering. Therefore, an optimization of the NIR sintering parameters was performed for this ink by introducing flash sintering. The influence of the distance between the NIR lamp and the sample on conductivity at different numbers of flashes is presented in Figure 8A. Preliminary test results (not shown) demonstrated that flashing and cooling times of 2 s yield optimal results. Once there is electrical conductivity, applying more flashes will not dramatically decrease sheet resistance which is again related to the decreasing emissivity and thus increasing NIR reflection. Increasing the lamp-to-sample distance leads to an exponential decrease of NIR radiation intensity on the ink´s surface if no temperature control is active. This explains the average increase in sheet resistance from 0.82 Ω/□ for a 0.5 cm sintering distance to an average value of 0.173 Ω/□ for 3 cm. Sintering at even longer source-sample distances results in a nonhomogeneous deposition, inducing a variable conductivity of the ink pattern.

The JS-A102A ink only requires two flashes at 0.5–1 cm distance to fully sinter the silver pattern. This corresponds with a total processing duration of 6 s at 8.3–9.8 W/cm^2^. Table 1 presents the maximum possible sintering intensities at different distances at 100% lamp power. The intensity at different distances is calculated based on the measured radiation angle of 120° and 100% efficiency. Compared with NIR sintering of JS-B40G, where 300 s of sintering is required, a spectacular reduction of 98% is achieved. Therefore, the findings of this study suggest that NIR flash sintering of JS-A102A is roll-to-roll compatible, in terms of processing time and conductivity. Figure 8B represents Ra surface roughness in nm for JS-A102A sintered samples as a function of source-substrate distance. In contrast to the NIR-continuous sintering of JS-B40G, increasing the sintering distance when NIR flash sintering of JS-A102A will lead to an exponential decrease of Ra roughness. Flash sintering at 3 cm creates very smooth, mirror-like silver layers with an average Ra surface roughness of 5 nm. At shorter distances (e.g., 0.5 and 1 cm) the NIR light becomes too intense given the sudden weight loss in the TGA profile upon heating, related to the sudden evolution of volatiles. For sintering distances greater than 2 cm, Ra roughness is less than 10 nm. However, an improvement in surface morphology is achieved, increasing the sintering distance is at the expense of the conductivity (Figure 8A), suggesting an incomplete decomposition of residual organic components in the ink. Thus, a trade-off between both figures of merit has been demonstrated.

Finally, in Figure 8C, the average layer and coffee ring thickness is presented. It can be observed that during NIR flash sintering, the average layer thickness and coffee ring peak thickness decreases linearly as a function of increasing source–substrate distance. This decrease is probably due to the speed of evaporation of solvents and the change of particle distribution. Sintering at longer distances results in less radiation density at fixed lamp intensity, therefore sintering is slower, and the particles have more time to migrate and settle. Flash sintering at 0.5 cm results in a coffee ring with a thickness of 137% as compared to the overall layer thickness, while flash sintering at 3 cm results in a coffee ring with a thickness of 150% as compared to the overall layer thickness.

Besides the influences of sintering conditions (such as distance and intensity) on the coffee ring effect, it is finally elucidated that ink formulation towards a specific sintering technique is crucial. As earlier described in the final part of Section 3.1, NIR-continuous sintering of an oven optimized ink leads to the formation of a prominent coffee ring due to the lack of balance in the convective flow mechanisms (capillary flow and Marangoni flow). The optimization towards this balance for a specific sintering technique becomes visible in the TGA analysis: one can observe that inks optimized for oven sintering consists of a complex solvent matrix ranging from low to high boiling solvents. This matrix ensures the above-mentioned balance during the relative slow sintering process. However, upon thermal analysis of the Flash NIR optimized ink, the solvent matrix proves to be more simplified since an optimized NIR ink is meant to fully sinter in matter of seconds, limiting the possibility of convective flows. Table 2 represents the coffee ring effect for different combinations of inks and sintering techniques. It can be concluded that it is very important to design the ink composition towards a preferred sintering method or vice versa.

## 4. Conclusions

For the JSB40G ink, a direct improvement of NIR-continuous sintering compared with oven sintering in terms of processing time is achieved. The sintering duration was reduced from 30–60 min to 2–5 min with no significant reduction in sheet resistance, which only slightly increases from 0.027–0.050 Ω/□ for oven sintering to 0.055–0.085 Ω/□ for NIR sintering. Furthermore, no significant increase of surface roughness, which slightly raises from ±13.5 nm for oven sintering to ±25 nm for NIR-continuous sintering, was found. However, the heating mechanism during NIR sintering causes an unbalance in the ink flow behavior (capillary flow and Marangoni flow), resulting in a very prominent coffee ring which can reduce the layer quality for several sensing applications.

It became clear that optimized inks can further reduce the sintering time down to several seconds. This optimization consists of three main modifications;
-Changing the particle size and distribution induces more absorption of the ink in the NIR spectrum, increasing the NIR sintering compatibility.-Adjusting the particle size distribution to have both small and large particles to facilitate Ostwald ripening.-Introducing more volatile solvents instead of a high boiling solvent, to reduce the required evaporation temperature and to block the convectional ink flows due to the reduced required sintering time.

NIR flash sintering of the JS-A102A ink reduces the sintering time down to 6–8 s together with the maximum processing temperature which dropped to 120 °C. Again, this method does not sacrifice on conductivity and surface roughness. The low boiling solvents used to optimize the JS-A102A ink for photonic sintering supports rapid evaporation and, hence, it reduces the coffee ring effect by more than half in comparison with the non-optimized JS-B40G ink as the possibility to create a capillary or Marangoni flow is limited. There is less time for the liquid to flow.

Finally, it is clearly shown for the first time that thermal imaging can actively follow the sintering quality since the emissivity measured with the device correlates with the ink track conductivity. To summarize, NIR sintering can be fully integrated into roll-to-roll machines and can be combined with other sintering methods, and thermal imaging can be implemented as an inline quality control unit to check the sintering coverage and quality.

## Figures and Tables

**Figure 1 nanomaterials-10-00892-f001:**
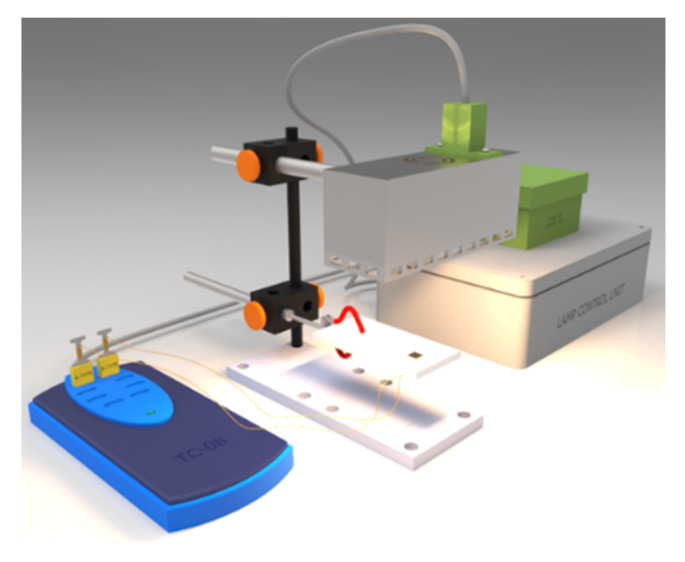
sintering setup. **Left**: Picolog thermocouple logger, **Center**: Near-infrared (NIR) lamp located above a Teflon (PTFE) substrate holder with two embedded thermocouples, **Right**: Control boxes with NI6008 USB IO device.

**Figure 2 nanomaterials-10-00892-f002:**
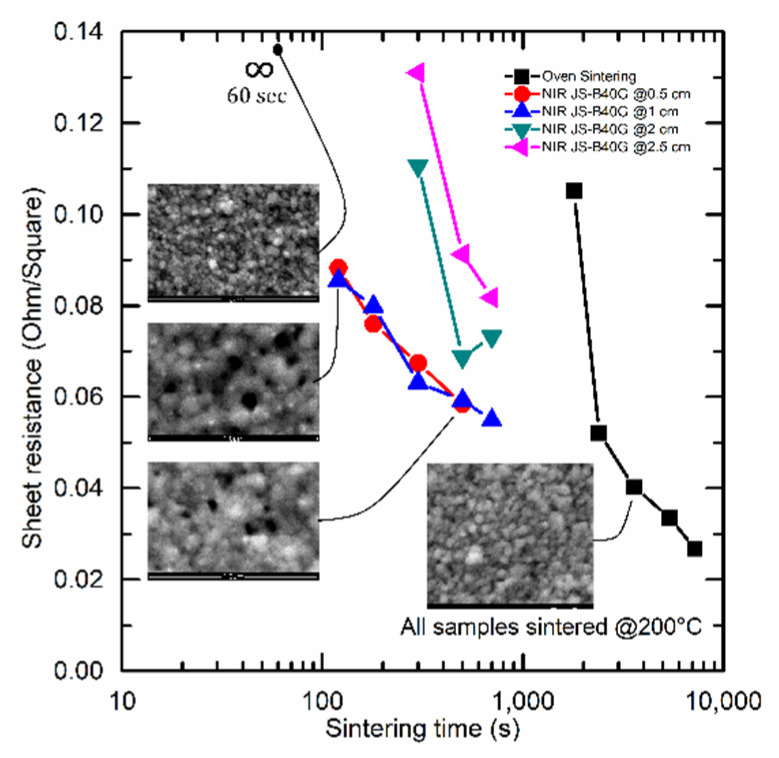
Sintering duration of near-infrared continuous sintering (blue-red-green-purple) and oven sintering (black) of JS-B40G ink. Switching from thermal to near-infrared sintering drastically reduces the sintering time, even with a non-optimized ink. Every scanning electron microscopy (SEM) image has a 1 µm field of view in horizontal direction, and shows the progression of particle growth from the early stage of sintering up to the final conductive layer.

**Figure 3 nanomaterials-10-00892-f003:**
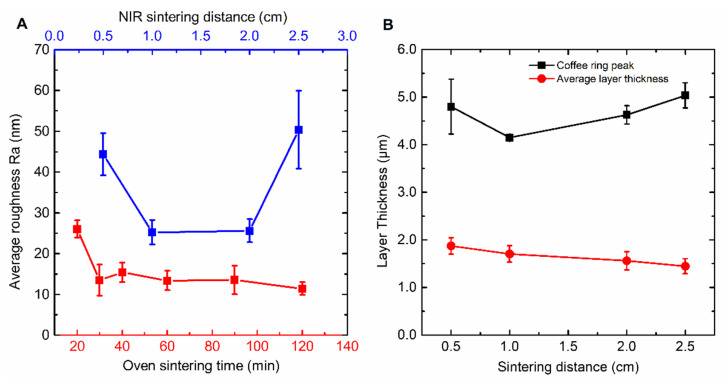
NIR-continuous sintering of JSB40G has an optimal sintering distance of 1 to 2 cm (blue), in order to achieve the lowest average surface roughness (Ra), with slightly higher results compared with conventional oven sintering (red). (**A**) In contrast to oven sintering, NIR sintering of JS-B40G results in the formation of a big coffee ring about two to five times the regular layer thickness. (**B**) Every datapoint consists out of three corresponding samples with every sample measured on two locations with the profilometer (line measurement), the error bars represent the standard deviation.

**Figure 4 nanomaterials-10-00892-f004:**
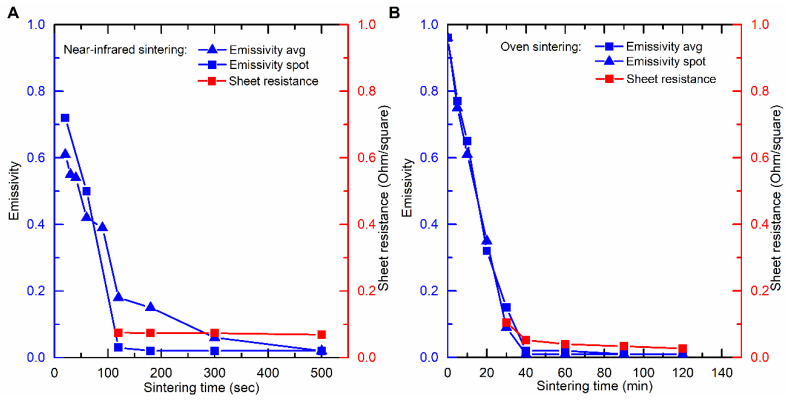
Average and spot emissivity during sintering correlate with the resulting sheet resistance of the ink layer for (**A**) NIR sintered JS-B40G ink and (**B**) oven sintered JS-B40G ink.

**Figure 5 nanomaterials-10-00892-f005:**
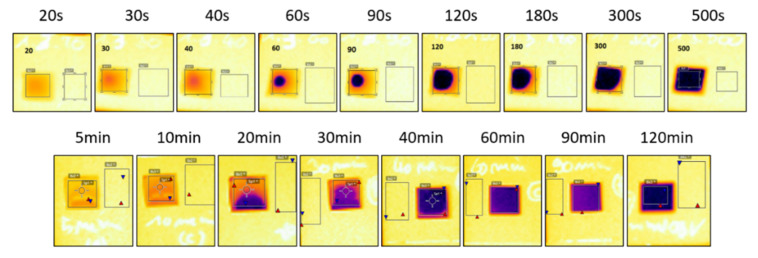
(E40) thermal images of the emissivity progression during sintering. Top row; NIR sintered JS-B40G. Bottom row; oven sintered JS-B40G. Dark areas have low emissivity; yellow areas have high emissivity.

**Figure 6 nanomaterials-10-00892-f006:**
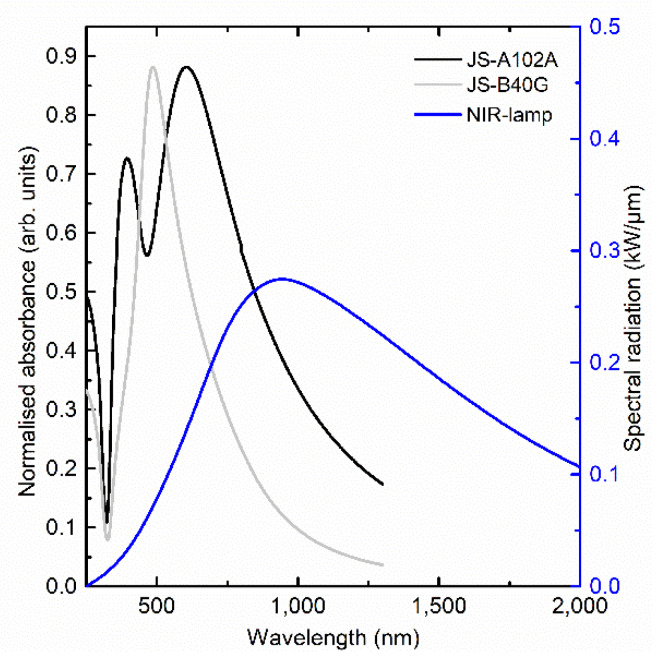
NIR compatibility of JS-B40G and JS-A102A ink. Blue: Specified spectral radiation (kW/µm) of the Adphos NIR light source. Grey: UV-Vis measurement, representing normalized absorbance of JS-B40G ink (light grey), having a low absorption overlap with the lamp and JS-A102A (dark grey), with a high absorption overlap with the lamp.

**Figure 7 nanomaterials-10-00892-f007:**
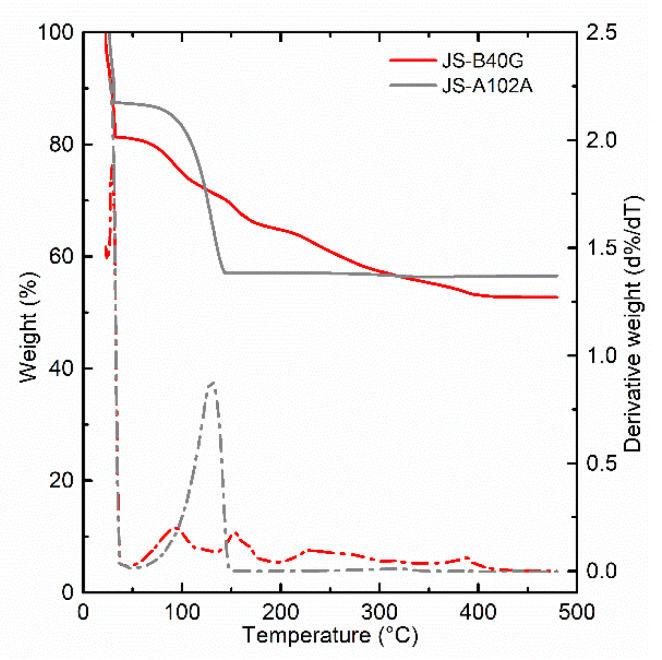
Measurement of both inks showing their relative weight and derivative weight. The weight loss (which correlates with the evaporation) of JS-A102A is very abrupt compared to the gradual loss of the non-NIR optimized JS-B40G ink, confirming the faster evaporation rate of JS-A102A. Secondly, the evaporation happens at lower temperatures.

**Figure 8 nanomaterials-10-00892-f008:**
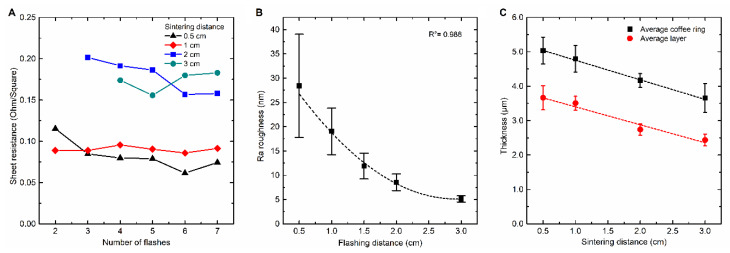
Sintering at shorter distances results in lower sheet resistance and requires a smaller number of flashes. (**A**) The higher light intensity at shorter distances causes a higher surface roughness, it exponentially decreases with increasing sintering distance according to the decrease of light intensity of Table 1 (Every datapoint consists out of three corresponding samples with every sample measured on two locations with the profilometer (line measurement), the error bars represent the standard deviation of each population.). (**B**) The layer thickness decreases linearly when increasing the sintering distance for JS-A102A at a flashing intensity of 100% and a flash and cooling time of 2 s (Each data point is representing the average of 8 to 12 measurements distributed over four to six different samples. The error bars represent the standard deviation of each population.). (**C**).

**Table 1 nanomaterials-10-00892-t001:** The theoretical maximum sintering intensity has an exponential decrease when increasing the lamp to sample distance. Based on a radiation angle of 120° and 100% lamp efficiency (@400W).

Sintering Distance (cm)	Maximum Lamp Intensity (W/cm^2^)
0.5	9.8
1	8.3
1.5	7.1
2	6.2
3	4.8

**Table 2 nanomaterials-10-00892-t002:** Overview of the appearance of a coffee ring during different ink and sintering configurations. Using the appropriate sintering technique according to the ink type leads to a significant lower coffee ring effect (130–150% instead of 240–350% of the average layer thickness).

Ink Type and Sintering Method	Coffee Ring Effect (% of Average Layer Thickness)	Ink Optimization
JS-B40G/OVEN	130–140	OVEN
JS-B40G/NIR (**cont.**)	240–350	OVEN
JS-A102A/OVEN	260–330	NIR
JS-A102A/NIR (**flash**)	137–150	NIR

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
