# Peer review of "Layer Morphology and Ink Compatibility of Silver Nanoparticle Inkjet Inks for Near-Infrared Sintering"

_nanomaterials, 2020, doi:10.3390/nano10050892_

Round 1
Reviewer 1 Report
This paper describes an extensive study of the comparison between oven sintering and NIR sintering. The introduction presents the context of the study and gives a detailed state-of-the-art on the different methods of sintering.
Then, the investigation covers the different questions arising from the use of NIR sintering : effet on the sheet resistance, compared to oven sintering results, effect of the distance of the NIR, influence of the sintering time, effect of the ink formulation... Several techniques are used to confirm the hypotheses the various steps of the study. The answers to the different questions are precisely discussed.
Only one comment on § 3.1 : the reason of the choice of the temperature (200°C) is not quite clear. This § could be improved. In addition, in the next §, the comment on Figure 2 mentions the "oven sintering (red)" whereas it is black on the scheme.
Author Response
Please see the attachment
(Note: due to the implementations of the comments of both reviewers, the line numbers in the letter will slightly vary from the original manuscript, the line numbers refer to the exact location after we implemented the comments.)

Reviewer 2 Report
This paper reports experimental findings suggesting that NIR sintering of Ag nanoink can be an attractive alternative to other methods like oven sintering. The findings made in this paper are interesting and can be of interest to engineers in the field. Since the aim of this research was for testing viability of NIR sintering, lack of scientific dissemination of data or discussion of mechanisms may be acceptable but not an ideal direction. Improvements in few areas may provide some level of enhancement in this aspect:
1) effect of NIR source to substrate distance effect: this is obviously related to the power intensity varying with the distance (as noted by authors). It is a good to have data but is a source of slight confusion. In the experimental section, the NIR setup was described to have an ability of adjusting intensity by the use of PID/thermometer. Yet, the test on the source to substrate distance effect sounds like that there exists some sort of mechanism affecting the sintering quality other than the source intensity. This needs to be revised to avoid such confusion;
2) providing better explanation as to the reason why NIR works better (shorter sintering time) even if the apparent temperature is kept the same as oven sintering. Local temperature (in between particles and particle surfaces) may be far higher than what was detected. This may be the reason why sintering can be achieved within such a short time given. In short, the mechanism as to how sintering is possible with far less time of exposure to heat seems necessary;
3) coffee ring effect: this is one of the most interesting observations, that is that NIR treated ink shows more coffee ring effect. This is potentially a "defect" of the NIR method and exploration as to its mechanism may make the paper more interesting and useful.
Author Response

(The authors gave the same response as above.)

Round 2
Reviewer 2 Report
The revised looks good.